# “Students Moving Together”, Tailored Exercise for Students Facing Mental Health Challenges—A Pilot Feasibility Study

**DOI:** 10.3390/ijerph20176639

**Published:** 2023-08-24

**Authors:** Kjersti Karoline Danielsen, Danielle Cabral, Silje Halvorsen Sveaas

**Affiliations:** 1Department of Nutrition and Public Health, Faculty of Health and Sport Sciences, University in Agder, 4604 Kristiansand, Norway; silje.h.sveaas@uia.no; 2Healthy Life Centre, 0578 Oslo, Norway; danielle.cabral@bgo.oslo.kommune.no

**Keywords:** mental health, students, physical activity, tailored exercise, physical fitness

## Abstract

An increasing number of university students are facing mental health challenges. The primary aim of this study was to determine the feasibility of 10 weeks of supervised tailored group exercise for 60 min twice a week delivered by the student health service for students facing mental health challenges. Secondary aims were to explore changes in mental health and physical fitness from pre- to post-test. Feasibility was assessed in terms of recruitment, drop-outs, attendance, and adverse events. The secondary outcomes included symptoms of depression and anxiety, wellbeing, satisfaction with life, cardiorespiratory fitness, and muscular endurance/strength. A total of 13 university students with self-reported mental health challenges, aged 20–39 years, were recruited during a four-week period. Ten (77%) of these completed the intervention and post-test as planned, and no adverse events occurred. There was a significant average reduction of 20% in symptoms of depression and anxiety (*p* = 0.008), and non-significant improvements of 21% in well-being and 16% in satisfaction with life were found. On average, cardiorespiratory fitness (*p* = 0.01) and muscular strength (push-ups test, *p* = 0.01, and sit-ups test, *p* = 0.02) increased. In conclusion, a 10-week tailored exercise intervention delivered by the student mental health service was found to be feasible, and beneficial for mental health and physical fitness in students facing mental health challenges.

## 1. Introduction

Worldwide, the prevalence and burden of mental health challenges and physical inactivity is increasing [1,2]. Facing mental health challenges affects the whole person and is reflected in thoughts, feelings, behavior, and social interactions with other people [1,3]. Mental health challenges are further associated with reduced psychosocial health and quality of life and are common reasons for sick leave and work disability, as well as dropping out of school and studies [3,4,5]. A national survey conducted regularly in Norway over the last decade shows a significant increase in the number of university students facing mental health challenges [6]. In addition, social restrictions during the corona pandemic led to a further increase in mental health challenges among students [6]. In 2021, one in three students reported that they faced mental health challenges and one in five students reported that they had a mental disorder [6]. However, knowledge regarding factors that may contribute to better mental health and quality of life for students is limited [7].

Given the limited capacity to assist everyone who seeks help for mental health challenges, there is a need for prevention and health promotion, as well as simple and easily accessible strategies students can use themselves to strengthen their mental health and better manage mental health challenges [5,8,9]. Physical activity shows potential in this respect. Physical activity is recognized as an important area in public health policy, and a positive association between physical activity and mental health has been found in a large amount of quantitative research [10,11,12,13]. In addition, several systematic reviews indicate that physical activity, in terms of exercise, can cause a moderate to large reduction in symptoms of depression, comparable to the effects of medical treatment and talking therapy [10,12,14,15,16]. In a recent randomized controlled trial, a 12-week group exercise program proved effective for patients with anxiety symptoms in primary care [17]. This is in accordance with several systematic reviews concluding that physical activity can be useful for people with panic disorders, generalized anxiety disorders, and social anxiety disorders [18,19,20,21]. Furthermore, physical activity can lead to improvements in symptoms for people with substance abuse disorders and psychotic disorders [22,23,24,25,26,27,28,29].

However, people facing mental health challenges may have barriers related to physical activity, and both quantitative and qualitative research emphasize the role of tailored exercise and facilitation via personnel with expertise in physical activity [30,31,32,33,34]. Nevertheless, research exploring the effects and experiences of tailored exercise for university students facing mental health challenges is sparse [35,36].

Mental health challenges arise in the context and society in which people live; thus, mental health recovery is expected to happen in psychosocial contexts in society [3,37]. For the students, this is at universities and colleges, and through activities on campus. Preventive and health-promoting strategies to promote students’ mental health, well-being and coping are cost-effective and simple ways to reduce the pressure on the healthcare system, making the students’ lives better and ensuring fewer serious mental health problems. Exercise can also be an important tool for improving educational achievements [38].

To summarize, there is solid evidence for the positive effects of physical activity on mental health and, to the best of our knowledge, there is a lack of clinical trials investigating the feasibility and effects of tailored supervised group exercise for students facing mental health challenges. Thus, the primary aim of this study was to determine the feasibility of 10 weeks of supervised tailored group exercise for 60 min twice a week delivered by the student health service for students facing mental health challenges. Second, we aimed to explore changes in mental health and physical fitness from pre- to post-test.

## 2. Materials and Methods

### 2.1. Study Design

This study was a singular group pre-post pilot feasibility study conducted in Southern Norway between October 2021 and February 2022 at the student health service located on the university campus. The study was a pilot test of an exercise intervention as a supplement to talk therapy for students seeking professional advice through the local student health service. Participation in the exercise intervention was separate from the students’ ordinary treatment. The study was approved by the Norwegian Center for Research Data (ref. 772348) and the Faculty of Health and Sports, University of Agder Ethics Committee (FEK). Reporting of the study follows the Consolidated Standards for Reporting Trials (CONSORT) 2010 statements [39,40], with adaptions to the non-randomized pilot and feasibility study design, and the template for intervention description and replication (TIDieR) checklist and guide [41] as appropriate.

### 2.2. Participants

The target population were students at the University and other higher education institutions. The inclusion criteria were students aged 20–39 years with self-reported mental health challenges, defined as challenges within the last 14 days, who were seeking professional advice through the local student health service. They were invited to participate in the study and received information about the experiment from employees when coming to talk therapy sessions. Exclusion criteria were pregnancy and students who could not attend the scheduled exercise sessions.

All participants received written and verbal information about the study, and they gave their written consent to participate in the study. They were guaranteed anonymity and informed that they could withdraw from the study at any time without any further consequences. Students who agreed to participate still received their normally scheduled talk therapy sessions with the employees at the local student health service alongside the intervention.

### 2.3. Sample Size

Guidelines for sample size calculation in feasibility studies have not been established [39,42,43,44,45]. Due to recruitment from a population known to have challenges related to participation in physical activity, and to ensure sufficient data to evaluate feasibility [45], we aimed to enroll 10–20 participants in the current study. A total of 13 students were enrolled in the study. Figure 1 provides a flow diagram of participants throughout the study.

### 2.4. Tailored Group Exercise Intervention

The intervention consisted of 10 weeks of tailored exercise in groups twice a week (a total of 19 exercise sessions). The overall goal was to present exercise possibilities for the participants and the experience of mastering and enjoying being physically active.

The exercise sessions were supervised by a master’s student in public health, who was an experienced personal trainer with competence in sports science and mental health and had five years of previous experience as a group instructor at fitness centers.

Most of the exercise sessions took place in a separate group training room at the university fitness center, except for four sessions that were carried out outdoors. In the first period (November–December 2022), one exercise session was carried out in the morning and one in the afternoon. In the second period (January–February 2022), both exercise sessions were carried out in the afternoon. During the intervention period, the personal trainer sent a personal message with a reminder to each participant prior to every exercise session.

The intervention included strength and aerobic exercises and focused on progression and variation. The exercise sessions lasted 60 min; 10 min warm up and cool down (dynamic stretching and aerobic exercises often in pairs) and 40 min of a combination of high intensity interval training and cross-training inspired exercises. The exercise method used was circuit training and AMRAPS (as many rounds as possible). At the beginning of the intervention, circuit training was used to ensure the correct technique and to become familiar with basic strength and conditioning exercises. At the end of the intervention period (approx. after three weeks), the instructor gradually increased the intensity of the sessions, and AMRAPS was used as an exercise method. Although the intensity of the sessions increased, mastery and enjoyment were still the main focus.

Although the exercise sessions took place in groups together with other students facing mental health challenges, the exercises were individually tailored to each student’s mental health, physical fitness, and function level. Alternative exercises were given to participants with special needs. To strengthen cohesion within the group, warm-up games and a collective cool down were carried out prior to and after each exercise session. A mid-term evaluation of the exercise program was conducted at the end of the first exercise period to ensure the participants’ motivation to continue the exercise intervention, and to receive feedback in regard to the exercise sessions.

### 2.5. Measurements

#### 2.5.1. Feasibility

**Recruitment of students**: evaluated through assessment of the recruitment process.

**Dropouts:** evaluated through quantification of how many students completed pre- and post-measurements.

**Adherence:** recorded by the personal trainer as the number of completed exercise sessions.

**Adverse events:** evaluated by the personal trainer as side effects or injuries caused by the exercise intervention.

#### 2.5.2. Mental Health and Physical Fitness

**Symptoms of depression and anxiety** were assessed using the Hopkins Symptoms Checklist-25 (HSCL-25), which contains 25 questions measuring symptoms of anxiety and depression. The scale for each question includes four categories of response (“Not at all”, “A little”, “Quite a bit”, “Extremely”, rated 1 to 4, respectively). The total score is the average of all 25 items, and a total score of >1.75 is regarded as in the range of high symptoms. HSCL-25 is often used as a measure for symptoms of mental illness and has been found to be both valid and reliable [46,47].

**Well-being** was assessed using the validated Warwick-Edinburgh well-being scale (WEMWBS) [48,49]. The WEMWBS includes 14 statements, and each is scored from 1 (“not at all”) to 5 (“all the time”), with a sum score from 5–70. Higher total scores indicate higher psychological well-being, and a sum score of ≤14 points is considered very low psychological well-being.

**Satisfaction with life** was assessed using the Satisfaction with life scale (SWLS) [50,51,52]. All items were answered using a 1–7-point scale, giving a total score from 5 to 35 points. A total score <9 points is referred to as low SWLS, and scores of 31 or over indicate high SWLS.

**Cardiorespiratory fitness** was measured using the 20-m shuttle run test for the estimation of VO_2max_ [53]. The test was performed with a five-minute standardized warm-up, which included a light jog (20 m marked distance), followed by max run for 15 s, then 15 s rest over a 10-min period. The 20-m shuttle run test has shown good reproducibility in other studies [54].

**Muscular endurance/strength** was measured through maximal repetitions of modified push-ups performed in 40 s, and maximal repetitions of sit-ups performed in 30 s. Push-ups and sit-ups tests have shown good results when measuring muscular upper body strength and are reliable for test-retest [55,56]. Maximal grip strength was measured with a grip strength test performed using a hydraulic dynamometer (peak force in kilograms). Three attempts were made, and the best results were used. This measurement has shown to be valid and reliable for maximal upper body strength, has shown to have good reproducibility for large and small samples, and is affordable and easy to use in a research setting’s retest [57].

### 2.6. Statistical Analysis

All statistical analyses were conducted using SPSS version 25 for Windows (IBM Corp, Armonk, NY, USA). All tests were two-sided, and statistical significance was accepted when *p* < 0.05. Data are reported as mean (standard deviation), 95% CI, or frequency (%). Baseline characteristics are shown for all participants, while only participants who completed pre- and post-tests were included in the analysis of change (n = 10). Testing for normality was done using the Shapiro-Wilk test and assessed by histograms and Q-Q plots. The changes in outcome measurements were analyzed with paired sample *t*-tests and Wilcoxon tests, as appropriate. There was little change in the *p*-values; hence, there was no change in the conclusion when using nonparametric versus parametric tests. Therefore, data are mainly reported as mean (standard deviation), and *p*-values from parametric tests. Intention to treat (ITT) analyses were also performed, where missing data were replaced by the last value carried forward approach, with no change in the results/conclusion.

## 3. Results

### 3.1. Sample Characteristics

A total of 13 university students (69% female) aged 20 to 39 years were included. The participants consisted of nine bachelor students (four in the first year, three in the second year, and two in the third year) and four second-year master’s students. The characteristics of the sample at baseline are shown in Table 1.

### 3.2. Feasibility Analyses

#### 3.2.1. Recruitment

Over the course of five weeks, the staff at the student health service identified 16 eligible students during scheduled talking therapy sessions and provided them with information about the study. They were further referred to the exercise intervention, and 13 of them agreed to participate. In the beginning, the employees forgot to focus on the possibility of recruitment. Thus, most of the students were recruited in the last two weeks when the employees remembered to recruit and had a positive attitude when they presented the exercise intervention for eligible students.

#### 3.2.2. Dropouts

As shown in Figure 1, 10 of 13 participants (76.9%) fulfilled the intervention and completed the post-tests, whereas three participants dropped out of the study. Two female participants dropped out after two exercise sessions, one of them due to social anxiety and the other due to compulsory lessons at the same time as the exercise sessions. One male participant dropped out after five weeks due to lack of motivation, disturbed circadian rhythm and being out of town the last five weeks of the intervention. The demographic and outcome’s variables at baseline did not differ significantly between the participants who completed the study (n = 10) and those who dropped out (n = 3). However, one of the dropouts had the highest score for symptoms of psychological distress (score 3.3) and the highest measured value on the fitness test (VO_2max_ of 49.9 mL/kg/min) at baseline.

#### 3.2.3. Attendance

Of the ten participants fulfilling the exercise intervention, one participant met the requirement for 80% attendance (which corresponds to 15 of the 19 exercise sessions). A total of eight (80%) had an attendance of ≥50% (i.e., 10 exercise sessions). Attendance for each participant is illustrated in Figure 2. Halfway through the study, it emerged that it was a challenge for the participants to attend morning sessions. This was also reflected in attendance, as there were clearly fewer participants attending morning sessions compared to afternoon sessions. It was then decided that all further sessions in the second half of the training period should take place in the afternoon.

#### 3.2.4. Adverse Events

One participant reported a sore knee during the intervention; accordingly, the exercise was adapted. Another participant vomited during the baseline aerobic fitness test and had to cancel; this participant returned the next day and completed the testing. No other adverse events or side effects were reported during or after the testing or the exercise intervention, indicating that safety concerns were adequately managed.

### 3.3. Preliminary Effectiveness

#### 3.3.1. Mental Health, Well-Being, and Satisfaction with Life

As shown in Table 2, there was a significant reduction in mean score for Hscl_25 from pre- to post-test, whereas the increases in SWLS and WEMWEB found from pre- to posttest were not significant.

As shown in Figure 3a, nine of ten students reduced their symptoms of depression and anxiety from pre- to post-test, and the number of students with a high Hscl_25 score was reduced from nine to six. Eight of ten students increased their well-being from pre- to post-test (Figure 3b); none of the students scored below the cut-off score (≤14) for low well-being either at pre- or post-test. Figure 3c shows that eight of ten students experienced an increase in satisfaction with life from pre- to post-test. None of the students scored above the cut-off score (>31) for high life satisfaction either at pre- or post-test.

#### 3.3.2. Physical Fitness

As shown in Table 3, there was a statistically significant improvement in cardiorespiratory fitness from pre- to post-test (4.7 mL/kg/min in estimated VO_2max_, *p* < 0.01). Furthermore, there was a significant increase in the mean number of repetitions in the modified push-ups and modified sit-ups tests, whereas there were no significant changes from pretest to posttest for grip strength.

Changes in physical fitness from pre- to post-test for each individual participant are shown in Figure 4a–d.

## 4. Discussion

This study examined the feasibility of 10 weeks of supervised tailored group exercise twice a week for students facing mental health challenges. Overall, our results indicate that it is possible to recruit and retain students facing mental health challenges for an exercise intervention. Only a few minor adverse events were reported; thus, it can be concluded that the intervention was safe for students facing mental health challenges. Furthermore, after the program, the students showed improvements in mental health and physical fitness. The promising results in this feasibility pilot study can be used for planning a larger and methodologically sound and robust clinical trial.

Despite a short recruitment period, the recruitment of participants for the study was fairly good for a feasibility pilot study [42,45,58,59]. However, it was highly dependent on the employees’ efforts, i.e., that they remembered to recruit and had a positive attitude when they presented the exercise intervention for eligible students.

Dropouts are a demonstrated challenge in exercise interventions for participants with mental health challenges [12,15,60], and the 23% dropout rate found in our study is consistent with previous exercise interventions for people facing mental health challenges [12,15,60]. In the current study, none of the subjects underwent the 19 exercise sessions planned, and we found lower attendance in the morning sessions. The low attendance in the morning exercise sessions in the first period may have influenced the results, which indicates the need for some changes in a future full-scale experimental study to improve the participants’ adherence to the intervention. Nevertheless, 80% of participants had an attendance of ≥50%. Previous studies have shown that people facing mental health challenges highly value facilitating personnel and a mutually supportive environment when exercising in groups with others in the same situation [30,31,32,33,34]. Hence, the social support from the other participants in the group and from the instructor during the exercise intervention in this study may be one of the reasons for good attendance. Furthermore, we believe that carrying out the exercise intervention at the university fitness center on campus may have had a positive impact on adherence, as it was close to the university and the living areas of the participants. However, qualitative research could establish the role of the group and location for attendance.

Second, we wanted to determine the impact of tailored group exercise on mental health and physical fitness. Conclusive inferences cannot be made given that this was a feasibility study without a control group. Still, promising findings included a significant decrease of 20% in symptoms of anxiety and depression after the intervention. This is in accordance with the positive effects of physical activity on symptoms of anxiety and depression in people facing mental health challenges shown in several systematic reviews and meta-analyses [10,13,14,15,16]. Furthermore, although not significant, there was an average increase in the students’ well-being and life satisfaction of 21% and 16%, respectively, after the intervention. Hence, this indicates that exercise has the potential to improve mental health, and that the outcome measurements were suitable in the detection of changes in mental health of this group.

Physical activity in the form of exercise is shown to have an equal or greater effect than traditional treatments, such as talk therapy and medication, for symptoms of depression [10,13,14,25]. However, the optimal type, intensity, duration, and frequency of exercise for depression has yet to be determined [12,61,62]. Some studies indicate that aerobic exercise and strength exercise have the same effect on reducing depression symptoms [12,61,62], and that a combination of endurance and strength training has a better effect than either endurance or strength training alone [62]. Furthermore, studies indicate that the effect appears to be greater for exercise with moderate and high intensity compared to low intensity [12,14,16,63]. In this study, the exercises and exercise dosages were individually adjusted and tailored to the participants’ physical and mental state by the instructor. Thus, the exact exercise dosage may be difficult to determine based on the current study and should always be individually adapted according to physical fitness and function level. In the current study, most of the participants attended 50% of the exercise sessions, and the results of this study showed that the participants had several positive changes in mental health and physical fitness. This may indicate that even a lower dose of physical activity than what was planned in our protocol may have a positive effect on mental health and physical fitness. This again supports the recent guidelines on physical activity, highlighting that all activity and every minute of activity have positive health effects regardless of length and intensity [64,65].

In the current study, we also found a significant increase in estimated VO_2maks_ and muscular strength, suggesting that the intervention was sufficient to improve physical fitness. If mental health challenges develop into severe mental illness, these conditions are associated with a 15–20 year reduction in life expectancy compared to the general population [66,67,68,69]. Although the causes are complex and partly unclear, a large proportion of excess mortality is linked to increased incidences of somatic diseases, such as cardiovascular disease, respiratory diseases, diabetes, obesity, and metabolic diseases [67,68,69,70,71]. For all of these are conditions, the effect of physical activity as both treatment, health promotion, and prevention are well documented [72]. In addition, low levels of physical activity and poor physical fitness are common among people with mental health disorders [11,73,74,75]. Therefore, introducing exercise as a simple and easily accessible strategy to strengthen mental health might be especially important for students in their early adulthood, as this may increase the likelihood of implementing exercise as a part of their everyday life in the future.

Mental health challenges often have an early onset [76,77,78], and a significant increase in the number of university students facing mental health challenges has been reported [6]. This emphasizes the importance of mental health promotion, prevention, and early interventions for this group. Therefore, simple and easily accessible strategies, which students can use to strengthen their mental health and better manage mental health challenges, are needed to support students’ future healthcare [7]. Tailored exercise may be an underutilized resource in the support and follow-up of students facing mental health challenges. In the current study, tailored group exercises showed promising results, thus indicating that tailored exercises may be an effective treatment for students facing mental health challenges. However, talking therapy will remain important for students facing mental health challenges, but it could be complemented by supervised group exercise, which is a preventative, health-promoting, and activity-based approach. Although no formal health economics analysis was undertaken, the cost of the interventions compares very favorably with one-to-one talking therapy, as there was only one personal trainer for the whole group two hours per week, and the cost-effectiveness of the exercise intervention should be further investigated.

Our results do not reveal what happens to the participants and their mental and physical fitness when they are no longer part of the exercise group. Whether participation in exercise continues is yet to be confirmed. The promising results of this feasibility study can be used for planning a methodologically sound and robust randomized controlled trial with a long-term follow up.

### Strengths and Limitations

The major strengths of the study are the employment of frequently used and validated questionnaires for mental health, which appear to capture changes and thus seem to be appropriate for a larger study. The fact that the exercise sessions were supervised and organized by an experienced and competent group instructor, who was responsible for all exercise sessions, is also considered a strength in this pilot study, as several studies emphasize the role and competence of the instructor as important in exercise interventions for people with mental health challenges [30,31,32,33,34].

The main limitation is the small sample size and lack of a control group. However, this was a pilot study, where the main aim was to explore feasibility in terms of the recruitment and delivery of the intervention [45]. Nevertheless, the sample size is sufficient to allow for the exploration of outcomes and may provide valuable and sufficient information for recruitment and power calculation when planning a future randomized controlled trial [45]. In addition, due to the population and the type of intervention in this study, a larger number of participants would have been inappropriate, as the group size of 10–20 students was a manageable group size for one training instructor. As the majority of the participants were female, the intervention should be further explored among male students facing mental health challenges.

## 5. Conclusions

The results of the current study indicate that 10 weeks of supervised tailored group exercise is feasible, acceptable, and safe for students facing mental health challenges, and positive changes were found in their mental health and physical fitness after the exercise intervention. The promising results support the idea that supervised tailored group exercise may play a role as a treatment for students facing mental health challenges, and thereby potentially alleviate stress on the healthcare system. The findings can inform future trials. To definitively determine treatment efficacy, future robust, randomized, and controlled trials are needed.

## Figures and Tables

**Figure 1 ijerph-20-06639-f001:**
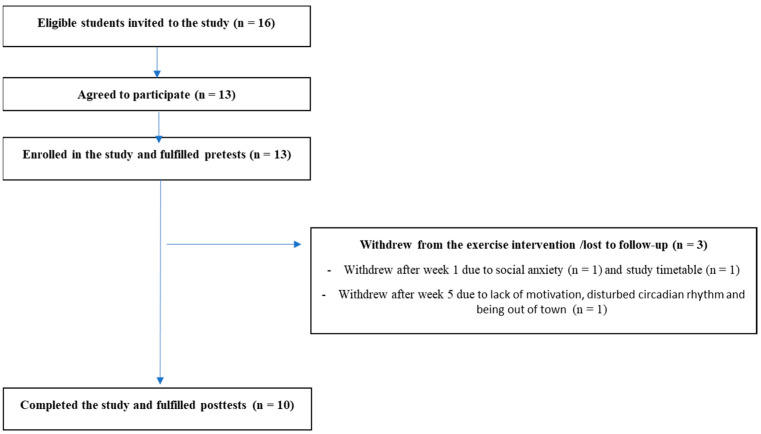
Flow of participants throughout the study.

**Figure 2 ijerph-20-06639-f002:**
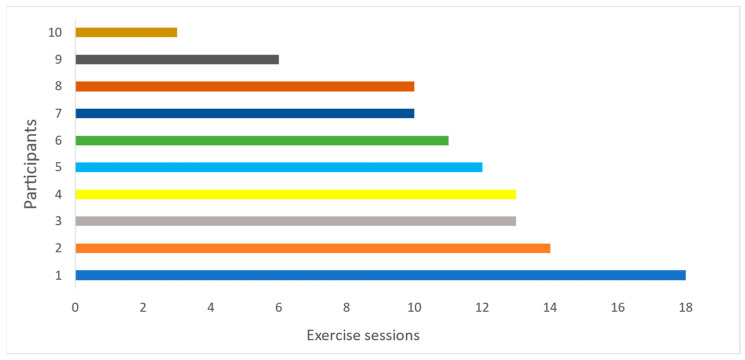
Attendance for each participant; each line illustrates one participant.

**Figure 3 ijerph-20-06639-f003:**
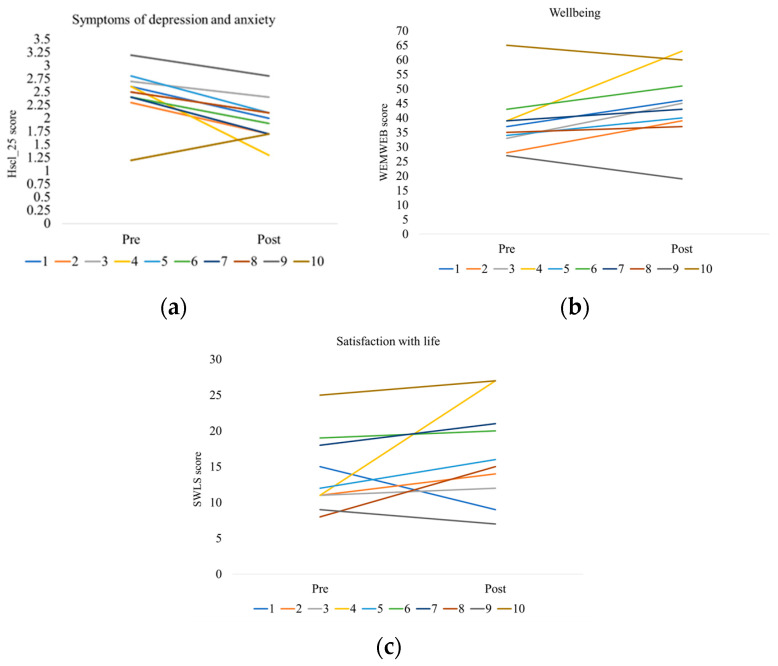
Change in mental health scores from pre- to post-test for each participant; each line illustrates one participant with the same color as in Figure 2. (**a**) Symptoms of depression and anxiety for each participant (HSCL-25, Hopkins checklist short version, scale 0–4); (**b**) Well-being for each participant (WEMWBS, Warwick-Edinburgh well-being scale, scale 5–70); (**c**) Satisfaction with life for each participant (SWLS, Satisfaction with life scale 5–35).

**Figure 4 ijerph-20-06639-f004:**
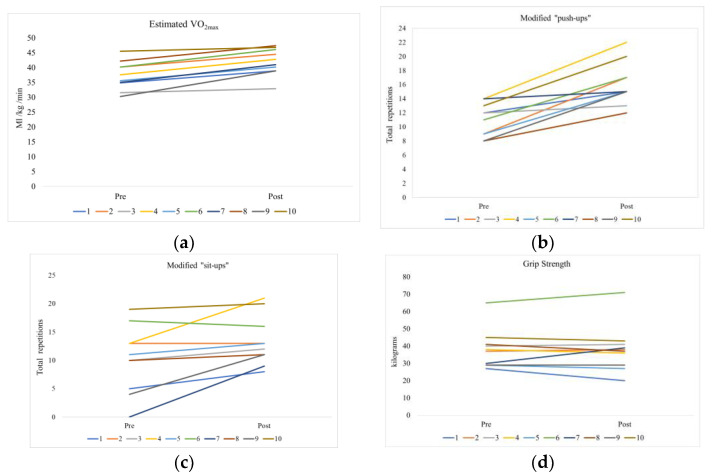
Change in physical fitness from pre- to post-test for each participant; each line illustrates one participant with the same color as Figure 2 and Figure 3. (**a**) Estimated VO_2max_ for each participant (20 m shuttle run test); (**b**) Maximal repetitions of modified push-ups for each participant; (**c**) Maximal repetitions of modified sit-ups for each participant; (**d**) Grip strength peak for each participant in force in kilograms using hydraulic dynamometer.

**Table 1 ijerph-20-06639-t001:** Characteristics of participants (n = 13) at baseline. Data are given as mean values (standard deviation), or numbers of participants (percentage).

**Demographics**	
Age, years, mean (SD)	24.5 (5.1)
Female, n (%)	9 (69.2%)
**Mental health**	
Symptoms of depression and anxiety	
Hscl_25, mean (SD)	2.5 (0.5)
Hscl_25 (<1.75), n (%)	12 (92.3%)
Well-being	
WEMWBS, mean (SD)	37.2 (9.4)
WEMWEB (≤14), n (%)	0 (0%)
Satisfaction with life	
SWLS, mean (SD)	13.3 (4.8)
SWLS (>31), n (%)	0 (0%)

HSCL-25, Hopkins checklist short version, scale 0–4; WEMWBS, Warwick-Edinburgh well-being scale, scale 5–70; SWLS, Satisfaction with life scale 5–35.

**Table 2 ijerph-20-06639-t002:** Changes in mental health (symptoms of depression and anxiety, well-being, and satisfaction with life) from pre- to post-test. Data are given as mean values (standard deviation or 95% confidence interval).

	Pre-Test (n = 10)	Post-Test (n = 10)	Mean Change (95% CI)	*p*-Value
Symptoms of depression and anxiety (Hscl_25)	2.5 (0.5)	1.9 (0.4)	0.5 (0.2, 0.8)	0.008 ^1^
Well-being (WEMWEB)	38.0 (10.7)	44.2 (12.3)	6.2 (−12.8, 0.3)	0.58 ^1^
Satisfaction with life (SWLS)	13.9 (5.3)	16.8 (6.9)	2.9 (−7.0, 1.2)	0.15 ^1^

^1^ = Paired Sample *t*-test. HSCL-25, Hopkins checklist short version, scale 0–4; WEMWBS, Warwick-Edinburgh well-being scale, scale 5–70; SWLS, Satisfaction with life scale 5–35.

**Table 3 ijerph-20-06639-t003:** Changes in physical fitness from pre- to post-test. Data are given as mean values (standard deviation or 95% confidence interval).

	Pre-Test (n = 10)	Post-Test (n = 10)	Mean Change (95% CI)	*p*-Value
**Cardiorespiratory fitness:**				
20 m shuttle run test (meters)	698 (94.5)	840 (94.3)	142 (95.1, 189.0)	<0.001 ^1^
Estimated VO_2maks_ (mL/kg/min)	37.3 (4.7)	42.0 (4.5)	4.7 (3.1, 6.2)	<0.001 ^1^
**Muscular endurance/strength:**				
Modified push-ups (max reps)	11.0 (2.4)	16.1 (3)	5.1 (3.2, 7.0)	0.01 ^1^
Sit-ups (max reps)	10.2 (5.9)	13.4 (4.4)	3.2 (0.7, 5.7)	0.02 ^1^
Grip strength (kg)	38.1 (11.2)	38.1 (13.6)	0.0 (−3.3, 3.3)	1.00 ^1^

^1^ = Paired Sample *t*-test. VO_2maks_, maximal oxygen consumption; max reps, maximal repetition.

## Data Availability

Data is unavailable due to privacy or ethical restrictions.

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
