# Peer review of "“Students Moving Together”, Tailored Exercise for Students Facing Mental Health Challenges—A Pilot Feasibility Study"

_ijerph, 2023, doi:10.3390/ijerph20176639_

Round 1
Reviewer 1 Report
Dear Editors,
The manuscript by Kjersti Karoline Danielsen, Danielle Cabral and Silje Halvorsen Sveaas submitted for publication in nt. J. Environ. Res. Public Health, entitled: “Students moving together”, tailored exercise for students facing mental health challenges.– a pilot feasibility study is a pilot study examined the feasibility of 10 weeks with supervised tailored group exercise twice a week for students facing mental health challenges.
Some comments and suggestions:
1. A reference to affiliation is missing next to author Silje Halvorsen Sveaas.
2. What is missing from the introduction is information on how the Covid-19 pandemic affected students' mental health, especially as the research was conducted during the pandemic, and this was a very important factor as many studies indicate.
3. The characteristics of the study group should be further developed. The authors only gave the age range. It is not even clear in which year of study the respondents were or what they studied.
4. It is not entirely clear how the subjects were invited to the experiment. Were the study subjects randomly selected or were they consecutively approached by the local student health service for advice?
5. Section 3.2.1 Recruitment from Results should be moved to Section 2.2 Recruitment and participants in Materials and Methods.
6. In the study protocol, the authors did not mention how many attempts of the grip strength test performed with the hydraulic dynamometer were made by the subjects. Most commonly, three attempts are used and an average is calculated.
7. The low attendance of the subjects in the morning exercise sessions in the first period of the study mentioned by the authors may have influenced the results of the study.
8. None of the subjects underwent the 19 exercise sessions planned in the study methodology by the authors.
9. The authors assert that the sample size was sufficiently large, an argument I am not convinced by.
10. Include the digital object identifier (DOI) for all references where available.
Kind regards
Minor editing of English language required.
Reviewer 2 Report
Reviewer Comment
Although the application of exercise intervention for the mental health development of university students is important and interesting as a subject, it is seen that there are many studies on this subject in the literature. In this sense, the difference, original value and importance of the study from the literature should be explained in more detail. What makes this study valuable should be known and the manuscript flow should be shaped accordingly. Apart from these, when the entire manuscript is examined, there are many places that do not comply with the article writing systematic. First of all, the information about the evaluation scales used should be added to the method part of the summary. In the introduction, it is seen that the integrity of meaning is not fully achieved. The transition between paragraphs and topics is very sharp. Conjunctions and related transitions should be made in terms of content. The study should be better justified, and at the end, the purpose should be written in line with the justification. The method section is written in a very complex way. Disclosures should be made in a more systematic way, as indicated in the Revisions section. The study design should be detailed. The content of the design should be moved to this section. The Participants section should be edited and the flowchart should be moved to this section. Evaluation scales should be gathered under the bond of measurements. The intervention protocol should be detailed and a special checklist should be used for interventions. For a more systematic writing of the manuscript, the article checklist should be used. Some information that should be in the method section is given in the result section. This information should be moved to the method section. The discussion section contained too many personal comments. The results should have been discussed in greater depth. It is seen that more space is allocated for information other than these. Discussion of the results should be prioritized. The study had many limitations, such as no control group, low number of participants, and no power analysis. The discussion should be organized in much more detail. Please make all revision suggestions in an orderly and diligent manner.
Revision
1. In the method section of the summary, please include the information on which assessment you made and with which scale. In addition, add the duration and frequency of the sessions to this section.
2. Page 1, line 22; “Symptoms of mental health challenges were significantly reduced (p=0,008)….” Fix the comma as a period. Make sure that the spellings you use are exactly the same as the manuscrit.
3. Page 1, 37-40; “National surveys in Norway over the last decade show a significant increase in the number of university students facing mental health challenges, and in 2021 one of three students reported that they faced mental health challenges and one of five students reported that they had a mental disorder [6].” If more than one study is mentioned in the paragraph, the sources of the studies mentioned as more than one should be at the end of the sentence.
4. Page 2, line 71-77; “Despite solid evidence for the positive effects of physical activity on mental health, to the best of our knowledge, no clinical trials have studied the feasibility and effects of tailored supervised exercise for students facing mental health challenges. The purpose of this study was to determine the feasibility of 10 weeks supervised tailored group exercise twice a week for students facing mental health challenges. The primary aim was to determine the trial feasibility in terms of recruitment, dropouts, attendance, and adverse events. Secondly, we aimed to determine the impact of the intervention on mental health and physical fitness.” As you mentioned at the beginning of this paragraph, if you are citing the literature, you must include the source. The statement that you have stated in this paragraph that there is no such study is a very clear and precise statement. Considering the possibility of you not being able to reach the study, you should use a statement that we could not reach the study. After these sentences, you directly stated the purpose of the study. The transition between sentences is done very independently. From the beginning to the end of the introduction, there should be a certain flow in the whole text, and all paragraphs and sentences should be connected with conjunctions to be related to each other. Conclude the introduction with the aim of the study by expressing the reason for the study, especially the purpose of the study. State the purpose more simply and clearly.
5. The material method section was very complex and not written in an orderly manner. Although you opened a topic called study design, you did not make any explanation. In the study design section, where and between which dates the study was conducted and a brief plan of the study should be stated. When the reader reads this section, he or she should be able to have information about the design of the study.
6. Follow the order in the material method section.
Study design: The plan of the study, information about ethical permission, when and where the study was conducted, and information about power analysis should be stated here. Also, use a checklist for manuscript writing. This checklist will allow the manuscript to be written in a more systematic way.
Participant: This section should specify the inclusion and exclusion criteria of the participants. Also, move the flowchart from the conclusion section here.
Measurements: All measurement methods should be specified here under this title. Please explain each evaluation method as a sub-title in this section.
Intervention: The intervention method should be described comprehensively in this section. You should also use an intervention checklist for standardization of the intervention method. “Template for Intervention Description and Replication (TIDieR) Checklist and Guide” is one of them. Use this or a similar checklist
7. Statistical analysis: In this section, explain how the normality test of the data is done. Then give information about whether the data are normally distributed and indicate which analysis method he used for which evaluation.
8. Page 2, line 92; In the intervention section, information on how many years of experience the person performing the intervention has in this field and what his/her area of expertise is should be added. In addition, the intervention is described very superficially. Information on which exercises are done should be given in detail.
9. Page 3, line 109; “2.4. Outcomes” Change this title to the measurements title and give each evaluation measurement as a sub-title more regularly.
10. Page 4, line 149; “2.5. Ethical Considerations and Procedures” Delete this header from the method section. Include ethical information in the study design.
11. Page 5, line 177; “3.2.1. Recruitment” Delete this section from the result section. The place where this information should be given is the material and method section.
12. Page 5, line 200 “Figure 1. Flow of participants throughout the study.” Move the flowchart to the participants section
13. Page 6, line 213; “3.2.4. Adverse events” The place where this information should be given is the participants part of the material method section. Delete this section here.
14. Page 6, line 212 “Figure 2. Attendance for each participant, each line illustrates one participant.” Move this figure to the material method section.
15. Please indicate which statistical method you use under the tables.
16. Although the numbers and some data in parentheses are given in the tables, the meaning of this information is not stated in the upper heading. Please add to the upper fish. (Ex: Main (SD))
17. Page 7, line 254; “Table 2. Changes in physical fitness from pre- to posttest. Data are given as mean values (standard 252 deviation).” Incorrect spelling has been used. Correct the table numbering as Table 3.
18. Page 8, line 264; In the discussion section, the article should be revised in accordance with the writing format in general. In the discussion, the main outcome data should be discussed in more depth and the inferences should not be personal but should be made by citing the literature. A more regular flow should be followed. In many parts of the discussion there are many paragraphs without reference. At the end of the discussion, the strengths and weaknesses of the study should be stated and the discussion should end with limitations. There are many limitations of your study, such as the lack of a control group, no power analysis, low number of participants, and non-homogeneous gender of the participants. Organize the entire discussion in this context.
19. Page 11, line 409; References should be arranged according to the writing system of the journal.
In general, there are places in the text that are inverted and where the integrity of meaning is not fully achieved. English regulations need to be done in much more detail.
Round 2
Reviewer 1 Report
Dear Authors
Thank you for your detailed response to the comments sent in the review and for making corrections to the manuscript.
Kind regards,
Author Response
Dear Reviewer.
We thank you once again for your valuable contribution to our manuscript. We could not find any comments from you to respond on this time. However, we have made small changes to the manuscript which we have highlighted using yellow marking of the text. In addition we have had a second proof reading of the paper and carefully edited the English language in the manuscript once again.
Kind regards, all authors.
Reviewer 2 Report
Reviewer Comment
Dear Author
It is seen that most of the revisions were made in line with the suggestions. However, a few more corrections are necessary for the manuscript to be in a better order. In particular, the comments should be cited in the discussion. In addition, the number of resources can be reduced too much. Please complete the editing suggestions completely.Tablo baÅŸlığında verdiÄŸiniz istatistiksel açıklamaları tablo altında dipnot olarak veriniz.
1. Inferences and comments in the Discussion section should be made by citing sources from the literature. Support your inferences with references from the literature. This indicates that your interpretation is based on evidence, away from personal interpretation.
2. Take the flowchart in the participants part of the method.
3. Please cite the intervention protocol.
4. “In addition, several systematic reviews indicate that physical activity, in terms of exercise, can cause a moderate to large reduction in symptoms of depression, comparable to the effects of medical treatment and talking therapy [10,12,14-22].” This explanation in the introduction does not need so many references. Too many resources have been used throughout the Manuscript. Reduce the source for this sentence and the entire manuscript.
Minor editing for the English version should make up for the shortcomings. In general, the manuscript has an understandable writing.
Author Response
Response to Reviewer 2 Comments
Manuscript IJERPH entitled “Students moving together”, tailored exercise for students facing mental health challenges. – a pilot feasibility study.
We thank you once again for your constructive comments to our manuscript. We acknowledge your work and have itemized, and point-by-point addressed your comments in the text below.
All our changes can easily be seen in the manuscript, since we have highlighted the changes using yellow marking of the text.
Point 1: Inferences and comments in the Discussion section should be made by citing sources from the literature. Support your inferences with references from the literature. This indicates that your interpretation is based on evidence, away from personal interpretation.
Response 1: As suggested more sources from the literature are cited in the inferences and comments in the discussion in line 306, 310, 340 and 401. We have also again reorganized some of the structure in the 3 first paragraphs, as well as reformulated sentences in line 303-304, 306, 317, 319, 322, 324-325 and 340.
Point 2: Take the flowchart in the participants part of the method.
Response 2: As suggested, we have moved the flowchart in the participants part of the method, and hence moved/changed the text in line 108-109 and 219.
Point 3: Please cite the intervention protocol.
Response 3: We assume you refer to citing a registration of the study protocol. As this was a small pilot study to explore feasibility of a further and larger clinical study, the protocol was not registered. We are now in the planning phase of a larger multi-center study, based on this pilot, which will be registered in Clinical.Trials.gov.
Point 4: “In addition, several systematic reviews indicate that physical activity, in terms of exercise, can cause a moderate to large reduction in symptoms of depression, comparable to the effects of medical treatment and talking therapy [10,12,14-22].” This explanation in the introduction does not need so many references. Too many resources have been used throughout the Manuscript. Reduce the source for this sentence and the entire manuscript.
Response 4: We agree, the oldest references are now removed, see changes in the introduction in line 52, as well as in the discussion in line 332 og 339.
Thank you once again for your valuable contribution, we have now also had a second proof reading of the paper and carefully edited the English language in the manuscript.
Kind regards, all authors.
